# Endocrine Aspects of ICU-Hospitalized COVID-19 Patients

**DOI:** 10.3390/jpm12101703

**Published:** 2022-10-12

**Authors:** Aristidis Diamantopoulos, Ioanna Dimopoulou, Panagiotis Mourelatos, Alice G Vassiliou, Dimitra-Argyro Vassiliadi, Anastasia Kotanidou, Ioannis Ilias

**Affiliations:** 1Department of Endocrinology, Diabetes and Metabolism, Evangelismos Hospital, GR-10676 Athens, Greece; 2First Department of Critical Care Medicine & Pulmonary Services, Medical School of National & Kapodistrian University of Athens, Evangelismos Hospital, GR-10676 Athens, Greece; 3Department of Endocrinology, Diabetes and Metabolism, Elena Venizelou Hospital, GR-11521 Athens, Greece

**Keywords:** critical illness, human, COVID-19, SARS-CoV-2, human, blood, plasma

## Abstract

The unprecedented scale of the current SARS-CoV-2/COVID-19 pandemic has led to an extensive—yet fragmented—assessment of its endocrine repercussions; in many reports, the endocrine aspects of COVID-19 are lumped together in intensive care unit (ICU) patients and non-ICU patients. In this brief review, we aimed to present endocrine alterations in ICU-hospitalized patients with COVID-19. There are tangible endocrine disturbances that may provide fertile ground for COVID-19, such as preexisting diabetes. Other endocrine disturbances accompany the disease and more particularly its severe forms. Up to the time of writing, no isolated robust endocrine/hormonal biomarkers for the prognosis of COVID-19 have been presented. Among those which may be easily available are admission glycemia, thyroid hormones, and maybe (OH)_25_-vitamin D3. Their overlap among patients with severe and less severe forms of COVID-19 may be considerable, so their levels may be indicative only. We have shown that insulin-like growth factor 1 may have prognostic value, but this is not a routine measurement. Possibly, as our current knowledge is expanding, the inclusion of selected routine endocrine/hormonal measurements into artificial intelligence/machine learning models may provide further information.

## 1. Introduction

The COVID-19 pandemic has taken healthcare providers by storm, particularly regarding critically ill patients who are hospitalized in intensive care units (ICU). During the earlier epidemic of severe acute respiratory syndrome (SARS), caused by the closely related SARS-CoV-1 virus, few studies evaluated some of this disease’s endocrine aspects. The unprecedented scale of the current SARS-CoV-2/COVID-19 pandemic has also led to an extensive—yet fragmented—assessment of its endocrine repercussions; in many reports, the endocrine aspects of COVID-19 are lumped together in ICU and non-ICU patients. In this brief report, we aimed to present endocrine alterations in ICU-hospitalized patients with COVID-19, including the results of our own research, and assess reported associations of tentative endocrine/hormonal biomarkers with disease severity and/or prognosis. Two literature search strategies were applied, on PubMed/PubMedCentral up to 15 August 2022. The first strategy included the following combination of keywords: ((ICU OR (critically AND ill) OR (critical AND care)) AND (SARS-CoV-1 OR SARS OR SARS-CoV-2 OR COVID-19) AND human AND hormone and produced 835 results. The second strategy included the following combination of keywords: (ICU OR (critically AND ill) OR (critical AND care)) AND (SARS-CoV-1 OR SARS OR SARS-CoV-2 OR COVID-19) AND human AND (hypothalamus OR pituitary OR thyroid OR adrenal OR gonads OR glucose OR diabetes) and produced 1890 results. The abstracts of these articles were scrutinized by two of the authors (ID and II); those articles honed on relevance for ICU patients were retained. Articles not relevant to ICU patients were excluded. Full-text versions in the English language were retrieved and were critically appraised for key results, limitations, suitability, quality, results obtained, and their interpretation and impact. Additional references were identified from the reference lists of the retrieved articles and were also appraised.

## 2. Non-COVID-19 Critical Illness

The central nervous system, the endocrine system, and the immune system work together to mount the body’s response to stress and harmful stimuli. Briefly, the endocrine changes of the acute phase of the response to stress involve the secretion of catecholamines, changes in the pulsatility of growth hormone (GH) secretion, an increase in cortisol, changes in thyroid hormones, the reduction in the concentration of sulfate dehydroepiandrosterone, the lowering of luteinizing hormone, an increase in prolactin, and increased resistance to insulin [1].

Critical illness entails considerable stress to patients; the body’s reaction includes an immediate increase in pituitary stress hormones, adrenocorticotropin (ACTH), GH, and prolactin (Figure 1).

In the acute phase of critical illness (days 1 to 7) mainly free triiodothyronine (FT3) decreases while in the chronic phase reduction in both thyrotropin (TSH) and free thyroxine (FT4) are observed. In the acute phase of critical illness, a dramatic increase in GH with a simultaneous decrease in insulin-like growth factor 1 (IGF-1) levels are noted. In the chronic phase of critical illness (after 7 days), central suppression of the growth axis is also observed, with a relative decrease in GH in addition to IGF-1. Testosterone is dramatically reduced and relatively elevated estradiol levels have been associated with a worse prognosis [2,3]. In the chronic phase, a central suppression of the hypothalamus-pituitary-gonadal axis (HPG) is also observed with a decrease in LH [2,3]. Prolactin increases in the acute phase and decreases in the chronic phase. In the acute phase, a significant increase in ACTH and cortisol is observed. In the chronic phase, a relative decrease in ACTH levels is observed whereas cortisol levels stabilize [4]. The acute reduction in binding proteins (TBG, CBG, SHBG, IGFPs) in critically ill patients affects the measurement of total hormones (cortisol, T3, T4, IGF-1, sex steroids) but also the action and clearance of their free moieties.

Of particular interest in critically ill patients is the hypothalamic-pituitary-adrenal axis (HPA). Accumulated evidence indicates that the noted increase in cortisol in such patients is in fact the result of low cortisol along with reduced cortisol breakdown, *in lieu of* increased cortisol production [5]. In critically ill patients, the expression of glucocorticoid receptors is diminished over time [6,7]. Pro-opiomelanocortin (the precursor of ACTH) is increased and may “whip” the adrenal cortex to produce some cortisol. In the chronic phase of critical illness, low ACTH, often due—in part—to the administration of corticosteroids, is associated with poor prognosis. A caveat regarding the assessment of adrenocortical reserve in ICU patients is that, despite earlier encouraging results, the cosyntropyn test may not be as reliable as previously thought in assessing adrenal reserve, due to the increased distribution volume of cortisol [5].

One-quarter of critically ill patients without prior history of diabetes have stress hyperglycemia [8]; the latter is attributed to activation of the HPA axis, increased insulin resistance, and release of proinflammatory cytokines [9].

Regarding prognosis, some endocrine parameters were associated with critical illness assessment tools’ scores, although there are caveats to be considered, especially as far as the HPA axis in general, and cortisol in particular, are concerned (please see above). Most studies have honed in on cortisol or thyroid hormones. Cortisol has been associated with the APACHE II tool score and the Glasgow Coma Scale score; the latter has also been associated with levels of thyroid hormones [9,10]. The prognostic value of adrenal androgens, prolactin, or gonadotropins has been explored to a limited extent. Prolactin has been correlated with the Glasgow Coma Scale, luteinizing hormone (LH) but not follicle-stimulating hormone (FSH) was associated with APACHE II, whereas adrenal androgens could be prognostic in septic shock [10].

## 3. Endocrine Aspects of SARS-CoV-1 Infection

Scant data in a few published reports have been put forth regarding endocrine aspects of the serious infection caused by SARS-CoV-1. Low TSH, T3, cortisol, and estradiol as well as elevated prolactin (PRL), FSH, and LH blood levels have been reported [11]. The thyroid was found to show extensive damage by the virus (with increased apoptosis and disturbed microarchitecture in follicular cells and absence of parafollicular cells) [12], whereas in the pituitary glands of five patients, scant positive cells and diminished staining intensity of immunoreactivity for GH, TSH, and ACTH were noted; the inverse was noted for PRL, FSH, and LH [13]. The relationship between SARS-CoV-1 and the adrenal glands is well-proven (the virus was found in the adrenal glands during autopsy studies) [14]. After recovery from infection, low ACTH and cortisol were noted in some patients [15]. Nevertheless, this finding could not be ruled out as being an after-effect of glucocorticoid administration during active SARS-CoV-1 disease. Glucocorticoid administration could also be implicated in cases of patients with femoral neck osteonecrosis. 

Acute onset diabetes/hyperglycemia was observed in half of the patients in a small series with SARS-CoV-1 infection; the ACE2-virus-attaching protein has been found in pancreatic islets [16]. Apparently, pancreatic islet cell function was compromised in the studied patients; in most of them normal glycemia was noted after three years of follow-up [16].

The endocrine aspects of patients with SARS are summarized in Table 1.

## 4. Endocrine Aspects of SARS-CoV-2 Infection

The thyroid gland and its function are among the most researched endocrine aspects of COVID-19 [17,18]. The ACE2 receptor for SARS-CoV-2 can be found in the thyroid gland [19]. In critically ill COVID-19 patients, low total T3 or free T3, low free T4, and low TSH have been found; these thyroid function parameters were noted to be associated with worse prognosis or mortality [20,21,22,23,24,25]. Researchers in Italy during the first “wave” of the pandemic reported laboratory and imaging findings that were compatible with the presence of transient subacute thyroiditis linked to COVID-19 [26]. Following this premise, we found that 60% of ICU COVID-19 patients and 36% of COVID-19 patients in the wards showed non-thyroidal illness syndrome (NTIS) and a thyrotoxicosis pattern as often in COVID-19 as in non-COVID-19 patients [27]. This NTIS pattern, albeit common, was related to the severity of critical illness rather than SARS-CoV-2 infection [27]. Tissue hypoxia is prominent in severe COVID-19; T3 may prevent this hypoxia and a relevant clinical trial has been initiated [28].

A high serum cortisol level (above 31 μg/dL; 855 nmol/L) upon ICU admission was reported to be the harbinger of a worse prognosis for COVID-19 patients [29]. This is in contrast to a study where cortisol was low in six of nine critically ill patients with COVID-19 [30]. Our team has found that the expression of glucocorticoid receptor alpha is increased in ICU hospitalized COVID-19 patients compared to non-COVID-19 cases [31].

Overall, the male sex lends susceptibility to more severe COVID-19; this is attributed to direct gender-hormonal status and indirectly to gender-associated behaviors (for instance women are more prone to seek medical help in case of illness compared to men) [32,33]. In women in the ICU with COVID-19 worse prognosis was reported when serum total testosterone was over 1 nmol/L [34]. Regarding men, analogous studies have yielded conflicting data: in one study total testosterone levels were higher in COVID-19 ICU patients [35], whereas in most other studies testosterone was lower compared to non-COVID-19 male ICU patients that served as control subjects [36,37,38]; in one study estradiol was also measured and was noted to be higher [36]. Bearing in mind that calculated free (FTe) and bioavailable (BavTe) testosterone may be more accurate measures of this hormone’s levels we measured total testosterone and calculated FTe/BavTe in 65 men with COVID-19 (of whom 10 were critically ill); all three measures were profoundly low, especially in the ICU patients and were associated with disease severity and worse prognosis [39]. 

Few studies have honed in on the growth axis in critically ill COVID-19 patients. Lower IGF-1 levels were found in ICU patients compared with mildly ill patients; the levels of IGF-1 were negatively associated with interleukin-6 (IL6) levels (which served as a measure of disease severity) [40]. In a study of ours, we found IGF-1 to be lower in critically ill patients who did not survive compared with survivors [41]. Interestingly, regarding the survival of critically ill patients with COVID-19, we found IGF-1 and GH to be comparable with the APACHE II and SOFA scores [41].

A field of intense study vis-à-vis COVID-19 is (OH)25-vitamin-D3 (VD3). Some studies have associated lower than normal and profoundly low VD3 levels with disease severity and prognosis since low levels are found in ICU-hospitalized patients with SARS-CoV-2 [42,43,44,45], whereas other studies have not found any association of VD3 levels with disease severity or mortality [46,47,48]. Furthermore, the current prevailing opinion is that VD3 signaling may be a part of the inflammation of COVID-19 but dosing on VD3 may not avert worse outcomes [49]. Nevertheless, maintenance of VD3 within normal limits is advised, in case of known nutritional deficiencies [50].

Patients with COVID-19 and pre-existing diabetes, compared with those without diabetes, are more prone to be hospitalized, be admitted to ICU/be mechanically ventilated, or even die [51]. Additionally, transient hyperglycemia has been noted in critically ill patients with COVID-19, even in the absence of a prior diagnosis of diabetes [51]. Recent research suggests that for many patients with limited access to health services, suffering from COVID-19 and exhibiting recent-onset hyperglycemia/diabetes, the infection apparently brings to the surface pre-existing undiagnosed diabetes [51]. Yet, fairly recently, we observed that in ICU-hospitalized patients with COVID-19 and no history of diabetes, beta cell function was overtly negatively affected [52]. Stress hyperglycemia in COVID-19 patients in the ICU may obscure the incidence of COVID-19-attributable hyperglycemia/new-onset diabetes [8]. Up to the time of drafting this work, the international registry of COVID-19-related diabetes established in 2020 (COVIDIAB; http://covidiab.e-dendrite.com/index.html, accessed on 25 August 2022) has not put forth any of its data [53]; it would be most interesting to see due to its international scale.

The endocrine aspects of ICU-hospitalized patients with COVID-19 are summarized in Table 2.

## 5. Conclusions

There are tangible endocrine disturbances that may provide fertile ground for COVID-19, such as preexisting diabetes. Other endocrine disturbances accompany the disease and more particularly its severe forms. The list of biomarkers for COVID-19 severity or prognosis is long and is growing [54]. To the best of our knowledge, up to the time of writing, no isolated robust endocrine/hormonal biomarkers for the prognosis of COVID-19 have been presented [54]. Among those which may be easily available are admission glycemia, thyroid hormones, and maybe VD3. Their overlap among patients with severe and less severe forms of COVID-19 may be considerable, so their levels may be indicative only. We have shown that IGF-1 may have prognostic value, but this is not a routine measurement. Possibly, as our current knowledge is expanding, the inclusion of selected routine endocrine/hormonal measurements into artificial intelligence/machine learning models may provide further information.

## Figures and Tables

**Figure 1 jpm-12-01703-f001:**
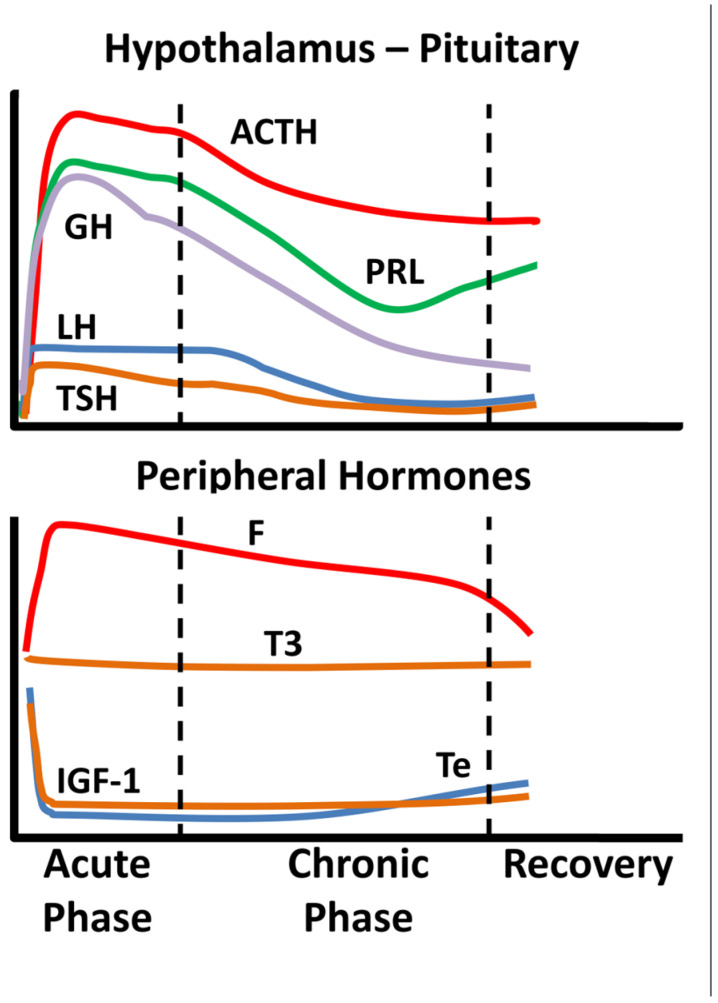
Schematic endocrine changes over time in critically ill patients; ACTH: adrenocortocotropin, GH: growth hormone, LH: luteinizing hormone, TSH: thyrotropin, PRL: prolactin, F: cortisol, T3: triiodothyronine, IGF-1: insulin-like growth factor 1, Te: testosterone.

**Table 1 jpm-12-01703-t001:** Summary of endocrine aspects of patients with SARS.

Affected Axis	
Hypothalamic–pituitary–thyroid	Low TSH, low T3 [11]; Extensive thyroid gland damage [12]; Diminished pituitary staining for TSH [13]
Hypothalamic–pituitary–growth	Diminished pituitary staining for GH [13]
Hypothalamic–pituitary–adrenal	Low F [11]; Diminished pituitary staining for ACTH [13]
Hypothalamic–pituitary–gonadal	Low E2 [11]; Elevated FSH, elevated LH [11]; Increased pituitary staining for FSH, LH [13]
Hypothalamo-prolactin	Elevated PRL [11]; Increased pituitary staining for PRL [13]
Glucose-insulin	Acute onset transient diabetes/hyperglycemia [16]

TSH: Thyroid Stimulating Hormone; T3: Triiodothyronine; F: Cortisol; ACTH: Adrenocorticotropic Hormone; E2: Estradiol; PRL: Prolactin; FSH: Follicle Stimulating Hormone; LH: Luteinizing Hormone; GH: Growth Hormone.

**Table 2 jpm-12-01703-t002:** Summary of endocrine aspects of ICU-hospitalized patients with COVID-19.

Affected Axis	
Hypothalamic–pituitary–thyroid	Low T3 or free T3, low free T4 and low TSH [20,21,22,23,24,25]; NTIS or Thyrotoxicosis [27]
Hypothalamic–pituitary–growth	Low IGF-1 [40,41]; Low IGF-1 and low GH associated with worse prognosis [41]
Hypothalamic–pituitary–adrenal	High [29] or low F [30]
Hypothalamic–pituitary–gonadal	In women with high Te: worse prognosis [34]; In men: high total Te [35] or lower Te than control patients [36,37,38]; low free Te and bioavailable Te [39], high E2 [36]
Glucose-insulin	Acute onset transient diabetes/hyperglycemia [51]; Impaired beta cell function [52]
Vitamin D	Low (OH)25-vitamin-D3 [42,43,44,45]

TSH: Thyroid Stimulating Hormone; T4: Thyroxine; T3: Triiodothyronine; NTIS: Non-thyroidal illness syndrome; F: Cortisol; E2: Estradiol; FSH: Follicle Stimulating Hormone; LH: Luteinizing Hormone; GH: Growth Hormone; IGF-1: Insulin like Growth factor-1; Te: Testosterone.

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
