# Peer review of "Endocrine Aspects of ICU-Hospitalized COVID-19 Patients"

_jpm, 2022, doi:10.3390/jpm12101703_

Round 1
Reviewer 1 Report
This invited concise review is well-written and deals with the main endocrine aspects of ICU-hospitalized COVID-19 patients. Minor remarks should be done:
-Report in the main text the rigorous methodology that you used to write your paper (i.e., search databases, languages of the selected articles, initial and final date of searching, inclusion and exclusion criteria to include/exclude the referrred papers.........)
-Create and add a Table where you can report the main findings on endocrine aspects and ICU-hospitalized COVID-19 patients to summarize your results.
-Add two best papers as references of your study: Scappaticcio L, Maiorino MI, Iorio S, et al. Thyroid surgery during the COVID-19 pandemic: results from a systematic review. J Endocrinol Invest. 2022;45(1):181-188. doi:10.1007/s40618-021-01641-1; Scappaticcio L, Pitoia F, Esposito K, Piccardo A, Trimboli P. Impact of COVID-19 on the thyroid gland: an update. Rev Endocr Metab Disord. 2021;22(4):803-815. doi:10.1007/s11154-020-09615-z.
Author Response
Reviewer 1 – Art. No: 1914125 “Endocrine aspects of ICU-hospitalized Covid-19 patients”
[1]. This invited concise review is well-written and deals with the main endocrine aspects of ICU-hospitalized COVID-19 patients.
We thank the Reviewer for his/her remarks.
Minor remarks should be done:
[2]. Report in the main text the rigorous methodology that you used to write your paper (i.e., search databases, languages of the selected articles, initial and final date of searching, inclusion and exclusion criteria to include/exclude the referred papers.........)
We thank the Reviewer for this suggestion. In the revised version of the manuscript we have added the following at the end of the Introduction section:
"Two literature search strategies were applied, on PubMed/PubMedCentral up to August 15th 2022. The first strategy included the following combination of keywords: ((ICU OR (critically AND ill) OR (critical AND care)) AND (SARS-COV-1 OR SARS OR SARS-COV-2 OR COVID-19) AND human AND hormone and produced 835 results. The second strategy included the following combination of keywords: (ICU OR (critically AND ill) OR (critical AND care)) AND (SARS-COV-1 OR SARS OR SARS-COV-2 OR COVID-19) AND human AND (hypothalamus OR pituitary OR thyroid OR adrenal OR gonads OR glucose OR diabetes) and produced 1890 results. The abstracts of these articles were scrutinized by two of the authors (ID and II); those articles honed on relevance for ICU patients were retained. Articles not relevant to ICU patients were excluded. Full-text versions in the English language were retrieved and were critically appraised for key results, limitations, suitability, quality, results obtained and their interpretation and impact. Additional references were identified from the reference lists of the retrieved articles and were also appraised."
[3]. Create and add a Table where you can report the main findings on endocrine aspects and ICU-hospitalized COVID-19 patients to summarize your results.
The Reviewer points to a shortcoming of our paper, which, hopefully, is rectified in the revised version of the manuscript. In this, we have added two tables, one for the endocrine manifestations of SARS-CoV-1/SARS and one for the endocrine manifestations of SARS-CoV-2/Covid-19. We believe that in this way the endocrine findings are conveyed more easily for the readers.
[4]. Add two best papers as references of your study: Scappaticcio L, Maiorino MI, Iorio S, et al. Thyroid surgery during the COVID-19 pandemic: results from a systematic review. J Endocrinol Invest. 2022;45(1):181-188. doi:10.1007/s40618-021-01641-1; Scappaticcio L, Pitoia F, Esposito K, Piccardo A, Trimboli P. Impact of COVID-19 on the thyroid gland: an update. Rev Endocr Metab Disord. 2021;22(4):803-815. doi:10.1007/s11154-020-09615-z.
We thank the Reviewer for the suggestion. These papers have been integrated in the revised version of the manuscript.

Reviewer 2 Report
I suggest you to present the endocrine aspects of SARS-Cov 1 and SARS-Cov 2 infection more clear, maybe organized on endocrine structure or secretion. You present a mixture of all, very difficult to follow and understand.
Author Response
Reviewer 2 – Art. No: 1914125 “Endocrine aspects of ICU-hospitalized Covid-19 patients”
[1]. I suggest you to present the endocrine aspects of SARS-Cov 1 and SARS-Cov 2 infection more clear, maybe organized on endocrine structure or secretion. You present a mixture of all, very difficult to follow and understand.
We thank the Reviewer for the suggestion. The Reviewer points to a shortcoming of our paper, which, hopefully, is rectified in the revised version of the manuscript. In this, we have added two tables, one for the endocrine manifestations of SARS-CoV-1/SARS and one for the endocrine manifestations of SARS-CoV-2/Covid-19. We believe that in this way the text is complementary to the tables and that the endocrine findings of SARS-CoV-1/SARS & SARS-CoV-2/Covid-19 are conveyed more easily for the readers.

Round 2
Reviewer 2 Report
The revised version of the manuscript is much better, in my opinion. The data are more clear for the readers, due to new introduced tables.